# The Advantages of Using Kaolin-Based Particle Films to Improve Coffee Production in the Minas Gerais Cerrado Biome

Newton de Matos Roda [1,*], Bruna Angela Branchi [1], Regina Márcia Longo [1], João Pontin [1], Deivisson Pelegrino de Abreu [2], Paulo Ricardo dos Santos [2,*] and Eliemar Campostrini [2]

1   Center for Economics and Administration, Campus I, Pontifical Catholic University of Campinas, Euryclides de Jesus Zerbini Street 1516, Campinas 13087-571, Brazil; bruna.branchi@puc-campinas.edu.br (B.A.B.); regina.longo@puc-campinas.edu.br (R.M.L.); jcpontin@gmail.com (J.P.)
2   Plant Physiology Sector, Center for Agricultural Sciences and Technologies, Northern Rio de Janeiro State University, Av. Alberto Lamego 2000, Campos dos Goytacazes 28013-602, Brazil; deivissonpabreu.uenf@gmail.com (D.P.d.A.); campostenator@gmail.com (E.C.)
*   Correspondence: newton.roda@tkinet.com (N.d.M.R.); prs_ufal@hotmail.com (P.R.d.S.)

**Abstract:** Climate change, such as increases in atmospheric air temperature, threatens Brazilian coffee production, which is mainly carried out on small rural properties in a family farming model. Increases in air temperature causes heat stress to the plants, resulting in physiological damage. This work studied the application of processed kaolinite-based particle films (PKPF) as part of environmentally sustainable agricultural practices. This innovative technology aims to increase the productivity of coffee grown in full sunlight and evaluate the interest of coffee growers in incorporating this input in the management of crops as an alternative to the traditional model of increasing production through the expansion of cultivated areas. This is a review of the state of the art of the use of PKPF in coffee, and a descriptive and exploratory research, supported by a literature review and field data collected, through a structured questionnaire applied to a group of coffee producers from the Cerrado in Minas Gerais. The concept of environmentally sustainable coffee production is latent among coffee growers, who adopt practical actions to respect and preserve the environment during the production process. Increases in productivity are related to the adoption of technological innovations such as the use of PKPF in the management of plantations, being a viable alternative to increasing the plantation areas, and thus reducing native vegetation.

**Keywords:** innovation; climate change; high temperatures; agriculture

## 1. Introduction

About 70% of the global coffee trade is derived from crops grown in tropical highlands, and Brazil stands out as a protagonist in the production of this *commodity*. Brazil supplies more than 35% of the total Arabica coffee (*Coffea arabica*) consumed annually in the world [1]. This species is native to the tropical forests of Ethiopia, located between latitudes 6° N and 9° N, at altitudes of 1600 m to 2000 m, with air temperatures ranging between 18 °C (minimum) and 22 °C (maximum). In these locations, this species grows and develops under the canopy of native trees, which have a direct influence on the growth and development of the coffee trees [2].

Environmental factors such as temperature, relative humidity, and solar radiation, combined with altitude and latitude, influence the regularity of the rainfall distribution regime [3]. The coffee plantations in Minas Gerais State (Brazil) are located at latitudes above 4° S, with tropical non-equatorial climatic conditions, and are therefore suitable for cultivation [4]. Climatic variables are crucial in defining the most suitable areas for Arabica coffee cultivation, and air temperature (average annual temperature) and the degree of water deficit during the critical periods of crop growth are fundamental for a high productivity [5].

The 2019 Special Report from the Intergovernmental Panel on Climate Change (IPCC) warned about the need for countries to make efforts so that the planet's temperature does not increase more than 1.5 °C in the next two decades. Changes in the climate regime cause impacts on crop productivity, and therefore future decision-making should take into account information related to climate change projections [6].

The Brazilian Cerrado is a very peculiar biome, as it comprises both open fields and dense forest formations. It predominantly occupies the Brazilian Central Plateau, with 206 million hectares, equivalent to about 23% of the national territory, and constitutes the second largest biome in the country, distributed mainly in the states of Minas Gerais, Goiás, Mato Grosso, Mato Grosso do Sul, Tocantins, Bahia, Piauí, Maranhão, São Paulo, and the Federal District. The Cerrado is characterized by having two well-defined seasons (dry winter and rainy summer), with an average annual rainfall of around 1500 mm. The dry period varies from four to seven months, and the rains are concentrated from October to March. The average temperature is around 22–27 °C, with the average maximum temperatures varying little over the months [1–3,6].

In the Cerrado region of Minas Gerais, during the period from 1974 to 2017, an increase in the mean annual temperature and a decrease in mean annual precipitation were observed. The effects of these changes in temperature and rainfall affect the physiological stages of flowering and ripening more drastically, which caused a reduction in productivity and a decrease in grain quality [7]. Due to climate change, commercial coffee growing in Minas Gerais is being challenged in order to maintain coffee productivity and quality, since studies of the plant's physiology have shown that temperatures above the ideal for cultivation can cause undesirable impacts, such as the degradation of chlorophyll molecules by excess solar radiation ("scalding", "sunburn") and a reduction in vegetative growth, with significant negative effects on productivity and a loss of bean quality [5,8].

Regarding the effects of temperature on coffee production, there would be an increase in air temperature of up to 5.8 °C associated with a 15% reduction in rainfall, and then the area suitable for coffee production would become limited to the mountainous regions located in the extreme south of Minas Gerais. This would cause huge socio-economic losses to communities that depend on coffee farming as a source of income, as well as having strong negative effects on the Brazilian economy [9].

The reports produced by World Coffee Research (WCR) [10] confirm that the increase in air temperature associated with changes in rainfall patterns interferes with the productivity and quality of the coffee beverage. Additionally, these changes contribute to increases in the number of insect-pests and the severity of plant diseases [11], which, by 2050, may affect production and the consequent reduction by half of the areas suitable for coffee growing [10,12]. Thus, it becomes important to consider all available agronomic practices, which are important for improving sustainability and to ensure the competitiveness of commercial crops which are vulnerable to stress factors, even if, among the few options, not all agronomic practices could be implemented immediately, due to availability and high costs demanded by these practices [13]. In summary, climate adversity and abiotic stress threaten Brazil's ability to ensure a regular supply of coffee for domestic and international markets. A conventional response would be to expand production into new areas, clearing forests and occupying pastures. However, this strategy may not be a long-term solution [14].

The challenges of agricultural sustainability demand a reflection, as well as a more holistic evaluation, aimed at preserving the already established social-ecological systems. Such action can be done through the interrelation of knowledge and practices [15], so that rural producers become increasingly engaged in the implementation of integrated and sustainable production systems, through the adoption of good agricultural and livestock production practices. Through this commitment, the ecosystem is preserved and the longevity of future production is ensured, in line with the 2030 Agenda and Sustainable Development Goals (SDGs), specifically the eradication of hunger (SDG 2), achieving gender equality (SDG 5), and reducing inequality (SDG 10). This Agenda proposes to end hunger, achieve food security, improve nutrition, and promote sustainable agriculture [16,17].

In this context of sustainable agriculture, the mineral kaolin $[Al_4Si_4O_{10}(OH)_8]$ processed, purified, and formulated for agricultural use, as a solar radiation reflective agent, reduces leaf, fruit, and canopy temperature, promoting beneficial changes in physiological and biochemical processes of the plant [13]. These benefits enable the plant to express its full genetic potential, with significant gains in productivity [8,18]. The use of this clay mineral can contribute greatly to environmentally sustainable management strategies for coffee growing, since it is an inert product and is widely used for organic agriculture in America, with its use certified by the Organic Materials Review Institute [19]. Thus, the objective of the current research was to study the use of kaolin as an environmentally sustainable product to increase coffee production in full sunlight, in the Cerrado biome of Minas Gerais, as an alternative to the conventional model that advocates increased production through the expansion of new plantations. Additionally, it was possible to evaluate crop management alternatives that increase the productivity of coffee plantations cultivated in full sunlight, in the face of climate change, notably a rise in temperature, and also to identify the degree of interest among coffee growers in including kaolin in the management system for coffee production in full sunlight, compared to cultivation in shaded areas by native or cultivated tree canopies. The work is based on descriptive and exploratory research [20], and the result of bibliographic analysis was combined with field investigation results, carried out through data collection with coffee growers located in the Cerrado Biome of the State of Minas Gerais, and through the application of a structured questionnaire.

## 2. State of the Art

### 2.1. Climate Change and Coffee Production around the World

In the different phenological phases of coffee plants, environmental factors can have severe impacts, which can compromise productivity and beverage quality. A large number of management strategies in coffee growing systems exist, and they can be implemented to avoid the effects of adverse weather conditions on the productivity and quality. Among these management strategies are the use of rootstocks adapted to abiotic stress, the selection of more tolerant genotypes, the proper management of the canopy to optimize the source–drain relationship, the management of water and soil use, and the application of special protective compounds [13,15].

For the period 2019 to 2030, growth rates have been predicted to range from 1.5% to 2.5% per year. Based on Brazil's significant participation in the global commercialization of this commodity, the Brazilian Agricultural Research Corporation (EMBRAPA) elaborated three strategic scenarios related to the global demand for coffee beans. If we look at one of the scenarios, i.e., the scenario in which the demand growth rate is fixed at 2% per year, the nations of the planet will consume approximately 208.80 million 60-kg bags of coffee in 2030. In order for Brazil to continue in its position as the largest coffee producer/exporter, and thus the most reliable global supplier, in 2030 Brazilian production must mandatorily increase by 74 million 60-kg bags [14].

Regarding the effects of climate change, several genetic improvement initiatives have been employed with the aim of increasing the tolerance of coffee plants to higher temperatures. However, these demand high resources and a long time period due to the complexity of the genetic structure of the species [20]. Coffee tree improvement, under abiotic stress conditions, for several traits simultaneously is laborious, complex, and costly. Thus, even considering the breeding strategies, biotechnological tools, and irrigation systems for a productive performance under heat stress conditions, the adoption of new sustainable alternatives in the short term is fundamental [21,22].

In the context of sustainable alternatives, the application of processed kaolinite-based particle films (PKPF) onto the leaves and fruits of plants grown in full sunlight can help plants to tolerate abiotic and biotic stress [8]. The use of PKPF as a stress mitigation agent, in particular high atmospheric air temperatures and excess solar radiation, has been intensively studied [18]. The efficiency has been proven, as the white protective film forms a physical barrier on the surface of the leaves, reflecting the excess solar radiation, promoting

the protection of the leaves and fruits against high temperatures. In the absence of this type of protection, the destruction of the leaf mesophyll may occur, with significant increases in the degradation of chlorophyll molecules [23]. The Marrakech Treaty, a result of discussions at the 22nd UN Climate Conference (COP22), reinforced the need for signatory countries to prioritize the implementation and acceleration of actions to reduce the rate of increase in global warming, and in this context Brazil plays a leading role in the preservation of natural ecosystems.

## 2.2. Brazilian Coffee Production in the Senario of Climate Adversity

The consolidation of Brazilian agribusiness, based on coffee farming, is strongly related to the negative effect of biotic and abiotic factors on this species [24]. The productivity, aroma, and flavor of coffee, as well as the intensity of pests and diseases, are related to variations in climate factors, particularly temperature and humidity [14]. The most suitable regions for growing high-quality and productive Arabica coffee are the mountainous regions, located at an altitude above 1200 m, with an average annual ambient temperature between 18 °C and 22 °C [25]. In Brazil, there are six coffee producing states, with Minas Gerais being the state with the highest production. This state accounts for more than 50% of the total coffee produced, and the coffee plants in Minas are grown over an area of more than 1 million ha. Thousands of farmers are employed in this activity [17].

In coffee plantations, studies on the impact of climate change indicate that the temperature is increasing by approximately 0.25 °C every decade. Concomitantly, in the periods of flowering and ripening of the beans, there has been a reduction in rainfall. This set of adverse weather events has caused a reduction in productivity of over 20%, particularly in the southeastern region, and especially in Minas Gerais [17]. Therefore, climate changes cause a reduced productivity and an increased economic and social vulnerability of coffee growers [7].

A study conducted at the experimental station of the Empresa Brasileira de Pesquisa Agropecuária de Minas Gerais (EPAMIG) [26] showed that the parts of the coffee plant exposed to excess solar radiation are compromised through damage to the photosynthetic machinery, resulting from the high degree of chlorophyll degradation [27]. Simultaneous to the reduction of the photosynthetic rate (photoinhibition), excess solar radiation raises the rates of transpiration, respiration, and photorespiration [18]. The degradation of chlorophyll molecules is observed through the presence of chlorotic spots, commonly known as "sunburn" or "scalds" [8].

Studies highlighting the sensitivity of coffee plants subjected to climatic adversity—specifically, instability in the rainfall and air temperature regime—have been carried out. Thus, more than 90% of the actual coffee plantations would be compromised if there is an increase of 6 °C in the current average air temperature [24]. There is evidence that climate change has already started and is already having a serious negative impact on coffee production and other crops of high commercial importance [28]. In 2014, the state of Minas Gerais, responsible for a quarter of the total production, experienced a period of severe drought, accompanied by high coffee prices, which motivated financial speculation [29].

Regarding the adverse effects of climate change on coffee plants, the physiological process and productivity of every coffee crop depend on the fertilization of flowers, fruiting, and the filling of the grains with water and photoassimilates. During the pre-flowering period, and also during the initial phase of coffee bean formation, the presence of low relative humidity and, simultaneously, temperatures between 20 °C and 23 °C can cause the dehydration of the initial floral buds, known as "floral abortion", and the fall of the "suckers". This phenomenon causes significant losses in coffee productivity [30].

## 2.3. The Use of PKPF in Commercial Agricultural Crops

Kaolin is a naturally occurring mineral, consisting of hydrated aluminum silicate, and the theoretical chemical composition is 39.50% aluminum oxide, 46.54% silicon dioxide, and 13.96% water. Naturally occurring kaolin has traces of undesirable metals, such as

red iron oxide ($Fe_2O_3$), which must be removed to achieve high quality, and also titanium dioxide ($TiO_2$), which must be eliminated to ensure safe handling and meet industrial specifications [31].

Kaolin is a versatile mineral, used in the cosmetics industry, in latex paints, as catalysts for oil refining, ceramic manufacturing, toothpaste, medicinal product formulations, food additives, and agricultural crop protection [32]. One of the most important uses of this mineral is for the bleaching of paper in the cellulose industries. It is chemically inert, has a soft texture, a high hygroscopic capacity, and a wide pH range. This mineral, commonly found in white color, does not undergo dilatation, is non-abrasive, and has low thermal and electrical conductivities [33].

Processed kaolin (PKPF) can be used as an ecologically acceptable agent in the management of commercial crops, because it is a product of low environmental impact and non-toxic. Mixed with water, and then sprayed onto plants, after the evaporation of the water, it creates a film of microscopic mineral particles adhering to the surface of the plant, be it leaf, branch, or fruit. Once on the plant surface, PKPF functions as a reflector of excess solar radiation and reduces the supra-optimal temperature stress. It has been verified that PKPF reduces the appearance of leaf burns (chlorophyll degradation and leaf mesophyll destruction), optimizes $CO_2$ assimilation, and improves productivity and fruit quality [34,35].

With respect to agricultural crops, to be effective, PKPF should have some of the following characteristics: having neutral pH, particle diameters smaller than 2 μm, formulations that enable uniformity of spreading and the formation of a uniform film on the plant surface. It should be porous enough not to interfere with leaf gas exchanges; allow the transmission of photosynthetically active radiation (PAR) to the leaf mesophyll; and also reflect ultraviolet (UV) and infrared radiation [8,34,35].

A PKPF product aimed at the agricultural market in Brazil, "Surround® WP", is commercially available, and its efficiency has been scientifically proven [7]. This commercial product is based on processed, purified, and formulated kaolin with characteristics suitable for agricultural use, which is under patent protection of Tessenderlo Kerley Inc. [36,37].

PKPF is classified as a non-hazardous, non-flammable, odorless agricultural agent that is free of any chemically synthetic ingredients. Therefore, given this set of characteristics, the PKPF is classified in Brazil as exempt from mandatory registration for commercialization, both by the Ministry of Agriculture, Livestock and Supply (MAPA), as well as by the National Health Surveillance Agency (ANVISA) and the Brazilian Institute of Environment and Renewable National Resources (IBAMA). These are the agencies which regulate all types of agricultural products in Brazil [38].

The parts of the plants exposed to full sunlight are protected after spraying the PKPF, when compared to plants without PKPF, with an average reduction of 4 °C, which reduced thermal stress, in addition to reducing the transpiration rate of the plants by up to 25% [13].

In olive trees it has been proven to increase the amount of oil extracted from harvested olives, as well as increase the oxidative stability and olive oil shelf life and reduce acidity [39,40].

In apple trees grown in New Zealand treated with kaolin, the fruits had less sunburn damage caused by excessive solar radiation on the leaves, and there was a 17% reduction in leaf temperature [41]. Similar results were observed in South Africa for apple trees that had been treated with PKPF, with fewer losses caused by leaf scald due to high solar radiation [42]. "Galaxy" apple trees grown in Turkey that received three PKPF spray applications had increased fruit quality and, even at postharvest, there was an increase in fruit mass, Brix, and titratable acidity [43].

In Chile, studies conducted with apple trees showed that PKPF sprayed on the trees three weeks before harvest reduced the surface temperature of the fruit by 1.5 °C and promoted a 70.1% increase in the photosynthetic carbon assimilation rate and a 61.3% increase in the stomatal conductance rate. This resulted in a 16% increase in productivity [44].

In the cultivation of vines intended for wine production, PKPF increased the concentrations of phenolic compounds in the berries, in addition to promoting an increase in

anthocyanins and ascorbate and greater antioxidant capacity [45–47]. As an additional benefit, the wine produced was visually more attractive and better appreciated when compared to wine produced from plants which had not been treated with PKPF [48].

In *Punica granatum* pomegranate plants, grown in India and in full sunlight, a 47% reduction in the rate of scalding of the fruit and leaf epidermis was observed once the surfaces had been treated with PKPF. The fruits free of blemishes on the peel had a better quality rating than those normally commercialized, and therefore PKPF applications have been considered as the best method for improving pomegranate quality in Turkey, as they have reduced sun scorch by up to 47% [49]. This agronomic practice has been incorporated into organic and integrated production systems of various crops in order to mitigate the effects of high temperatures in orchards [48,50].

In Egypt, when considering commercial tomato cultivation, a reduction in leaf temperature by 1.1 °C was observed, and there was a significant reduction in scald symptoms in the tomatoes exposed to full sunlight. As a consequence, there was a reduction in losses and improved quality in the marketed produce [31].

In Italy, another study conducted with tomato plants showed that in the fruit in which the epidermis received PKPF application, the epidermis was free of sunburn spots, and the fruit was classified as of "superior quality". As such, fruits had a higher added commercial value, which resulted in the generation of an additional profit of up to €900 per cultivated hectare [51]. The additional benefit promoted by PKPF, was the repellent action to insects called sharpshooters (Hemiptera: Cicadellidae), which transmit the bacterium *Xylella fastidiosa*. This bacterium is responsible for causing a lethal disease in orange, mango, tangerine, lemon, apple, grape, and coffee orchards [52,53].

Recent studies conducted in Brazil have shown that spraying PKPF on orange plants contributes to repelling pests called psyllids, *Diaphorina citri*, by up to 80%. This pest, feared by many citrus growers, acts as a vector for the transmission of the bacteria that causes greening (Huanglongbing/HLB), one of the most destructive diseases, and with the power to decimate citrus orchards of all ages in Brazil and worldwide [54,55].

### 2.4. PKPF Efficiency in Coffee Crop Management in Brazil

Photosynthesis is a process of complex chemical reactions, which depends on both adequate photosynthetically active radiation (PAR) and optimal leaf temperature for maximum $CO_2$ assimilation [55]. Research conducted between the 1950s and 1970s had shown that coffee plants are sensitive to ambient temperature greater than 25 °C, with the photosynthesis process coming to a halt at 34 °C [56]. Commercial coffee crops grown in full sunlight, which received PKPF spraying, were evaluated under unfavorable weather conditions and proved that the plants were more productive when compared to plants without PKPF protection [57]. Coffee trees exposed to temperatures above 23 °C, without forest shading, tend to present a higher rate of flower abortion, inducing the plant to produce more leaves and fewer berries [8,58]. Research evaluating the effect of PKPF on the sensorial quality of the coffee beverage, conducted on the Catuaí Vermelho IAC-44 variety, showed that a plantation without PKPF treatment recorded 71.66 points, compared to 77.13 points obtained in the area where PKPF was sprayed onto the plants, an improvement of 9.2%. The biggest benefit of the increased sensory score was that the selling price of the coffee was 15.5% higher than the price of a 60-kg bag of the standard treatment coffee harvested by the farmer [59]. Another scientific study published by the same authors showed a 28.4% increase in grain quality and that each kg of grain-dried coffee beans with PKPF protection yielded 0.635 kg compared to 0.594 kg from the area harvested without application of this product. Coffee plants protected with Surround® WP produced 94 60-kg bags per hectare, compared to 91 bags from plants without protection [60]. Studies of the protective capacity of PKPF for the leaves of young *C. arabica* and *C. canephora* coffee plants, transplanted from the nursery to the field, in full sunlight, over two seasons of the year (autumn and summer), demonstrated that PKPF had a protective effect on the leaves of both coffee species, keeping

the leaves healthy, vigorous, and free of symptoms of sunburn, when compared to plants which had not treated with PKPF [8].

## 3. Materials and Methods

A descriptive study, which contemplated the observation of the facts recorded and analyzed without any form of manipulation by the researcher, was carried out with an exploratory characteristic, considering that the objective was to obtain more information on the theme related to the use of PKPF technology by coffee producers [20,61].

As for the technical procedure of data collection, the research was classified as bibliographic and field research [31–62]. It was divided into two stages, the first being a data search, to substantiate the theme of use of PKPF as a reflective agent to reduce excess solar radiation, with an emphasis on the use in coffee production. In a second step, field research was conducted by means of sixteen individual interviews, in 2021, amongst a previously mapped universe of sixty-five coffee growers, who cultivate Arabica coffee in full sunlight on their properties, located at an altitude of between 900 m and 1150 m above the sea level, ideal for coffee cultivation, within the Cerrado Biome of the State of Minas Gerais, the second largest biome in South America [26].

The sample was intentional, also known as "judgment sample", and is part of the non-probability sampling group. Therefore, it demands a greater participation and involvement of the researcher in choosing the elements of the population that make up the sample [63]. The choice of this type of sampling is based on the fact that PKPF was recently introduced into the Brazilian market [8], and the number of farmers who use this product is limited, compared to the universe of existing coffee growers in Brazil [64].

## 4. Results

The first important result was that the majority of the farmers interviewed, 94%, are aware that they are responsible for preserving the Cerrado biome, where they play the role of protagonists in the preservation of the environment and where they have extracted, for more than two decades, sustenance for their families. In an open question, they recognized that "Sustainability is the use of natural resources, such as soil and river water, to produce coffee and how important it is to preserve the biome for their descendants". These farmers also impact the social fabric of the municipalities where they grow coffee, by generating jobs and reducing hunger and inequality.

About 81.2% of the coffee growers interviewed mentioned that they incorporate residues from cultivated crops and/or animal excrements such as chicken and cattle manure, seeking to fully replace, or at least reduce as much as possible, the use of chemical fertilizers for coffee cultivation. Sixty-eight percent revealed that they select and use products of biological or natural origin, environmentally safe, less polluting, and less toxicologically aggressive to humans, which are used for crop protection against pest insects and fungal and/or bacterial diseases. These pests/diseases cause severe economic losses. Table 1 shows the interviewees' answers.

**Table 1.** Environmental sustainability practices adopted by farmers in the Cerrado region of Minas Gerais.

| Sustainable Management Practices and Plant Protection in Coffee Crops in the Cerrado of Minas Gerais—Brasil | |
| --- | --- |
| Sustainable management practices | 1 *, 5, 7, 8, 9, 10, 11, 12, 14, and 16: use of residues (coffee straw) to replace soluble fertilizers; use of chicken manure as organic fertilizer; reuse of coffee residues to produce fertilizers (compost); use of animal waste and wishes (urine and grated barnyard manure) as fertilizers |
| Sustainable practices plant protection | 2, 3, 4, 6, 8, 12, and 15: integrated pest and disease control; "...I take care when dosing the products in the (spraying) tanks to avoid water contamination"; "...I always try to use organic products"; "... I use biological product.."; I don't use herbicides, just brush cutters"; Search for alternatives that are less toxic for the environment |

* Interviewee.

The interviewed coffee producers associated the concept of sustainability with the compliance with Federal Law 9.974/2000, regulated by Decree 4.074/2002, which, among other requirements, defines the principles for the environmentally correct disposal and handling of empty pesticide containers. This action is part of the activity of the production chain, whose responsibility must be shared among all agricultural production agents, including farmers and the government.

Throughout the interviews, all coffee growers mentioned being aware of the importance of the disposal and environmentally correct disposal of empty containers of pesticides used annually in coffee plantations. Only 25% of them made a point of recording in their answer that they strictly followed the recommendations of the National Institute for Empty Packaging Processing (INPEV), and considered this practice as an effective way of preserving the environment. Therefore, it can be inferred that all of the farmers involved in this research contribute to a sustainable coffee production system, since they are active agents of the reverse logistics, whose focus is on the recycling of empty containers, labels, and leaflets. Thus, pesticides become inert and harmless to the environment, since the packaging, when transformed into raw material, feeds back into the production chain [65].

Sixty-nine percent of the interviewed coffee growers recognize that increasing production by expanding into new areas may cause some degree of impact on the environment, and 31% admitted that there are very high negative impacts (Figure 1A).

Considering plantations already in the productive period and the use of PKPF for more than two consecutive harvests, 75% of coffee growers declare that they perceived the possibility of increasing the production of coffee through additional investment in PKPF, used as an input to mitigate the effects of high temperatures as a substitute for shading the coffee with the native tree canopy, and affirm that this input does not cause any impact on the environment (Figure 1B). The remaining 25% did not know how to evaluate this impact. It is important to emphasize that all the interviewees have included PKPF in the management of their plantations for two or more harvests.

The IPCC Panel report shows that there will be more frequent and more intense extreme environmental events, notably prolonged droughts, heat waves, storms, and hurricanes [66–68]. In view of this information, it is relevant to understand the perception of coffee growers about what will be the main expected climate threats, for which 75% of the interviewees mentioned temperature change, and 25% emphasize the reduction in frequency and volume of rainfall.

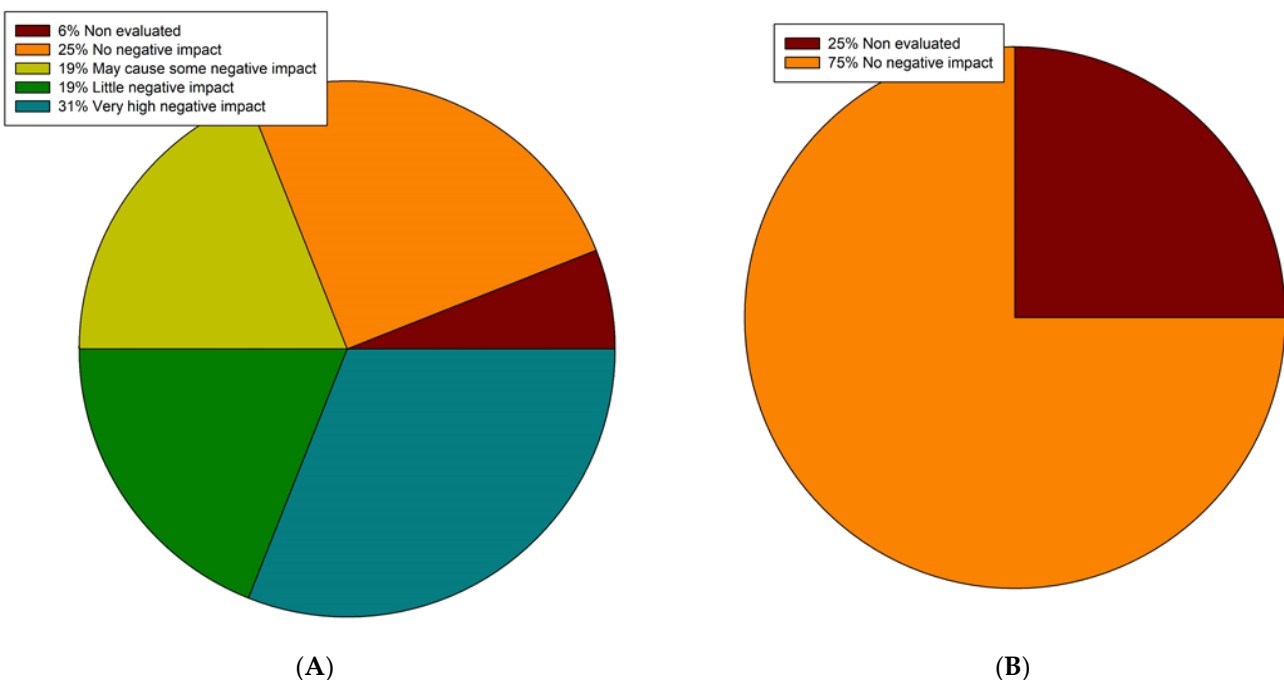

(**A**)  (**B**)

**Figure 1.** (**A**) Evaluation of the environmental impact of leasing new planting areas; (**B**) Evaluation of the environmental impact of using kaolin to promote increased coffee production without increasing the planted area.

These answers and statements, consistent with the temperature increase cited in the IPCC report [67], are supported by the average temperatures registered in the region. Figure 2 summarizes the evolution of the average temperature in one of the municipalities of the coffee-growing region in Minas Gerais, which registered an increase in the average air temperature over the period between 1980 and 2008, during the spring, summer, and fall seasons.

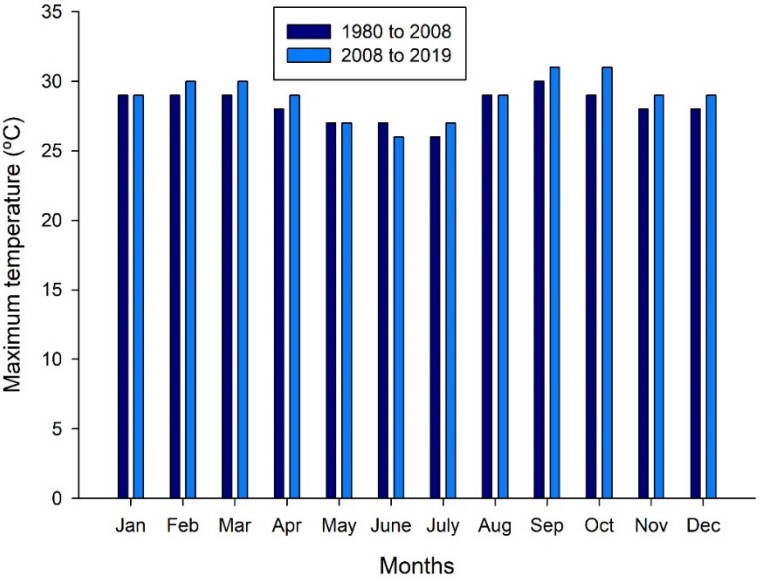

**Figure 2.** Average atmospheric temperature recorded per month, over the periods 1980–2008 and 2008–2019 in the municipality of Patos de Minas/MG.

Although genetic improvement programs have developed Arabica coffee varieties that are more tolerant to higher temperatures [68,69], the rise in average temperatures above 0.5 °C in most months of the year triggers an important alert, especially during the month

of February, with an increase of 0.9 °C precisely in the flowering period of the plant, and in the month of October, with an increase of 1.8 °C above the historical average (Figure 2).

The importance of the increase in air temperature on the productivity of coffee plantations was understood by the interviewees (Figure 3A). All interviewees recognized that temperatures above 30 °C would cause a significant impact on coffee yield.

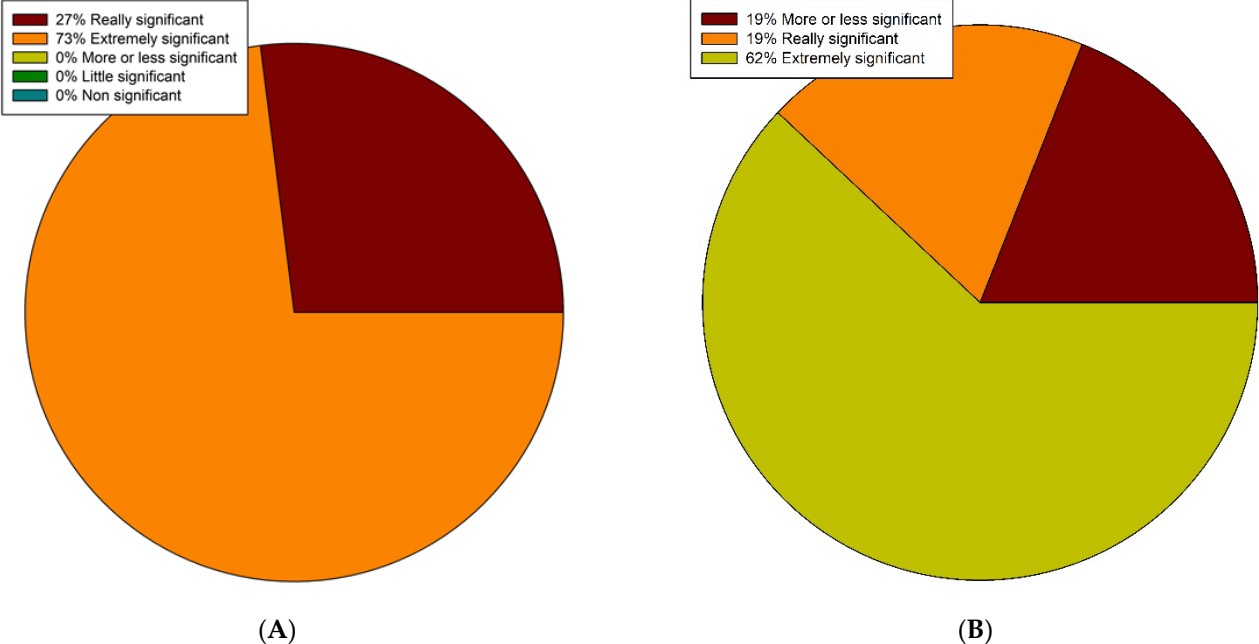

(**A**)                                                                                               (**B**)

**Figure 3.** (**A**) Evaluation of the impact of atmospheric temperatures above 30 °C on the productivity of harvested coffee (60-kg bags); (**B**) Evaluation of the reduction in the quality of coffee beans harvested with atmospheric temperatures above 30 °C.

In addition to a reduction in productivity (Figure 3B), of even greater concern is the reduction in quality of the coffee beans harvested when average recorded temperatures are equal to or higher than 30 °C. All the interviewed coffee growers recognize that there is a loss of quality, and 81% reported that this information was very or extremely important (Figure 3B). The market demands for better quality coffee are responsible for the diffusion and adoption of new technologies for coffee production and preparation. Among the specialty coffees, organic coffee is one of the most prominent in this segment, and, as reported in the literature, the selling price of a bag of coffee is linked to the production system, as well as the quality of the beans [8,70].

Concomitantly to climate change, there is a change in the regularity and volume of rainfall [34]. The adoption of methods such as irrigation helps to reduce the impacts caused by increases in temperature. However, besides the high cost of implementing irrigation, this management technique is dependent on a regular and sufficient source of water [34,71]. Figure 4A summarizes the importance that the interviewed coffee growers attributed to irrigation as a management technique to increase productivity and mitigate the variability of rainfall, considering that an absolute majority, 87%, adopted the drip irrigation system, and thus have guaranteed the water supply and ensured the productive longevity of the crops.

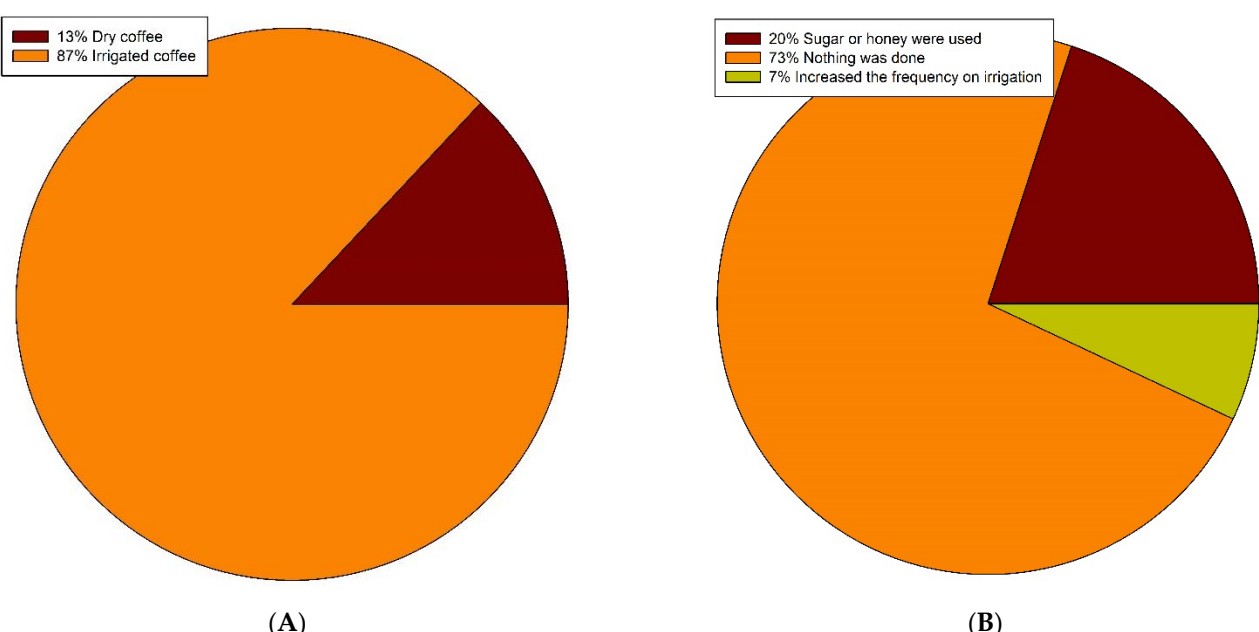

**Figure 4.** (**A**) Distribution of total cultivated area by type of coffee production management; (**B**) Production management actions to reduce the impacts of high atmospheric temperatures.

The relevance of the impact of temperature increase on the physiological process of coffee plants was recognized by the interviewees and corroborates the evidence found that on hot, sunny days, especially in summer, when there is an incidence of the sun's rays directly on the surface of the leaves and coffee beans, resulting in more frequent scalds [8,15].

Most of the coffee growers (73%) reported that, before learning about the option of using PKPF, they did nothing to protect the crop from excessive solar radiation and high temperatures (Figure 4B). Only 27% of them performed some kind of action to mitigate the negative impacts caused by high temperatures, of which 20% said they used sugar or molasses, empirically, and 7% increased the irrigation frequency (Figure 4B).

Legitimized by the fact that all the interviewees knew and used PKPF (Surround WP®) in the management of coffee production, in at least two consecutive harvests, the survey revealed that 69% of the farmers recognized PKPF's performance as "excellent", 25% evaluated it as "very good", and 6% classified the input as "good" for the protection of the leaves and berries, according to their experience (Figure 5A).

Regarding the effects of the PKPF as a reflective agent of solar radiation, able to reduce the water stress and thermal stress of coffee plant surfaces during periods of high air temperatures [8,70], 53% of the interviewees, the performance of PKPF was excellent, while 33% rated it as "very good", and 7% rated it as "good" (Figure 5B).

Knowing that supra-optimal temperatures affect the productive performance of plants, the survey investigated the actions taken by farmers to increase production in the face of observed climate change (Figure 3). Among the seven options offered, 94% of the interviewees recognized as "extremely important" and "very important" the inclusion of PKPF in the current management of coffee plantations, by spraying the plants about two or more times within a production cycle. Furthermore, 88% of the farmers considered it "very important" and "extremely important" to combine the spraying of PKPF with foliar fertilizer, within the same spray tank, in order to increase productivity.

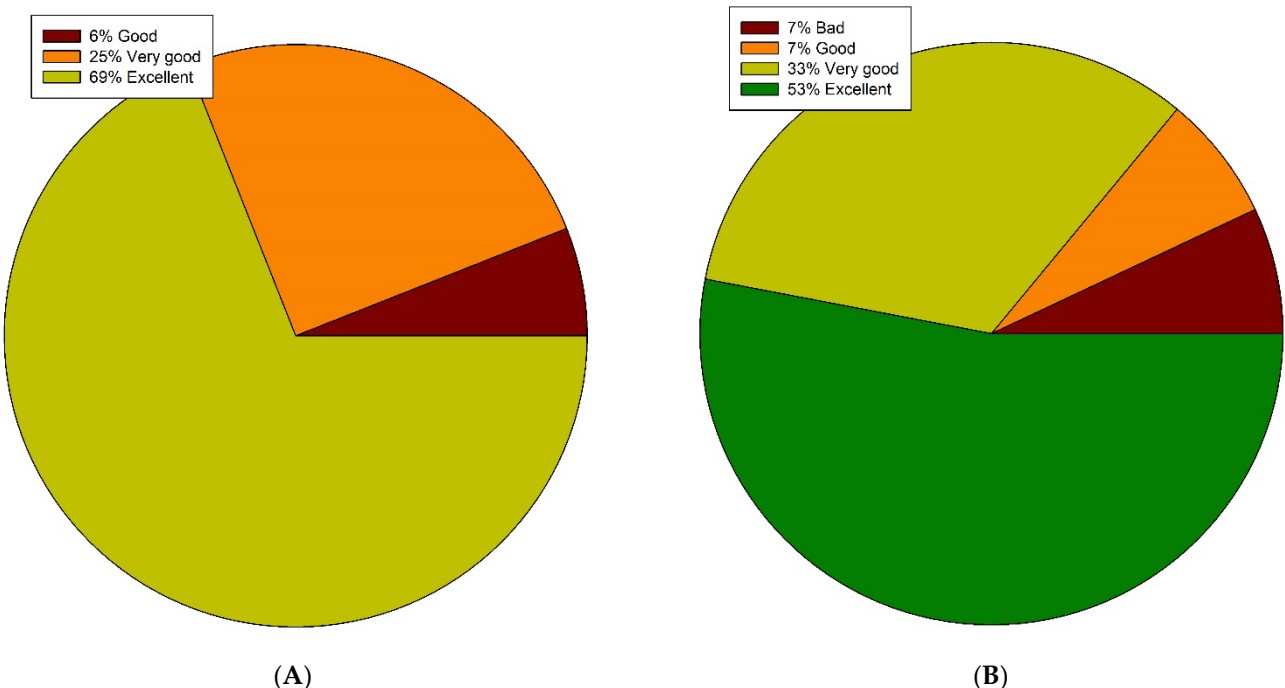

(**A**)                                                                      (**B**)

**Figure 5.** (**A**) Evaluation of PKFP as a scald protector of leaves at an average atmospheric temperature above 30 °C; (**B**) Evaluation of PKFP as a scald protector on beans with an average atmospheric temperature above 30 °C.

Regarding the other strategies to increase productivity, almost all of the interviewees (93%) believed that increasing the dose of soil fertilizer contributed to increasing productivity, given the low rates of natural fertility and marked soil acidity, typical of the Cerrado biome [71] (Figure 6).

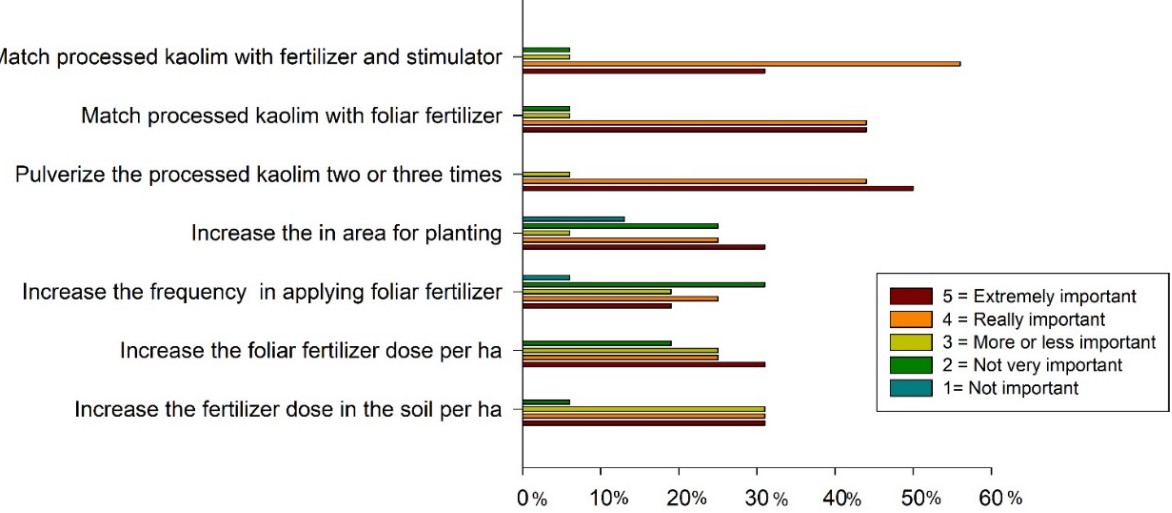

**Figure 6.** Evaluation of options for increasing coffee production.

Looking at these figures, more than half of the farmers recognize that it is important to increase the cultivated area. This contrasts with their role as "environmental preservation protagonists". We can write something like this: Differently than expected, more than half of the farmers indicated that increasing the planting area was an extremely (31%) and very (25%) important action to increase production.

Unlike the concept of production, the concept of agricultural productivity is a relative measure of the amount produced per unit of cultivated area [70,71]. With the aim of

increasing productivity, there is a viable trajectory for increased production, without the need to expand to new planting areas [71].

The interviewees, all with more than two decades of experience in coffee plantations, recognized as decisive factors for productivity irrigation (93.8%), high temperatures and the use of fertilizer (81.3%), and the presence of insect pests in the soil and/or aerial parts of the plant (75%), and only 68.8% recognized the importance of diseases of fungal and/or bacterial origin. These results confirmed the concern with the increase in air temperature, and that temperatures above the optimum can compromise the growth and development of the coffee tree (Figure 7).

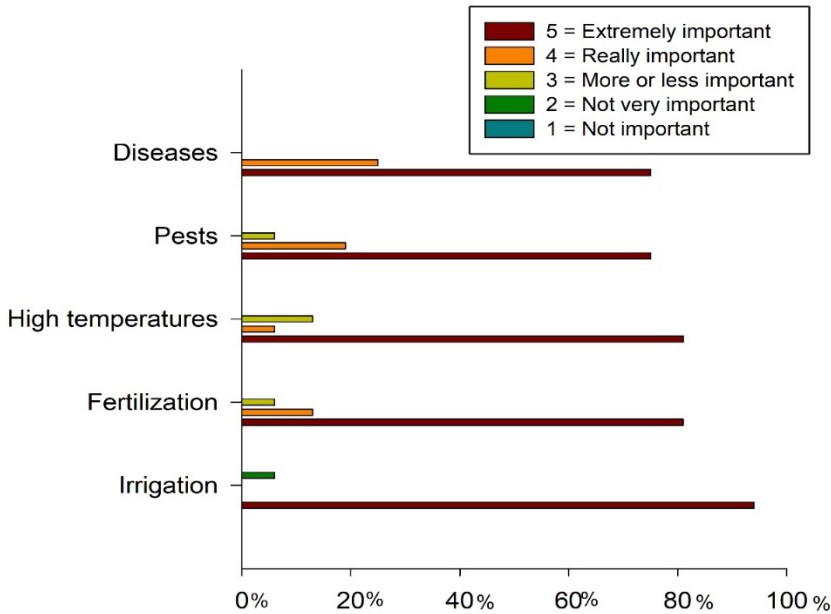

**Figure 7.** Evaluation of the importance of factors that interfere with productivity.

In a general evaluation of the different effects of PKPF on coffee crops, 88% of the interviewees stated that it was "extremely important" or "very important" to use PKPF to increase yields and improve plant vigor. Eighty-one percent pointed out that PKPF protects and improves the quality of the berries until harvest, and 76% reported that the product ensures a greater uniformity of ripening. Regarding the repellent action against pest insects, such as the leaf miner or the coffee berry borer, 25% of the producers considered it to be moderately important (Figure 8).

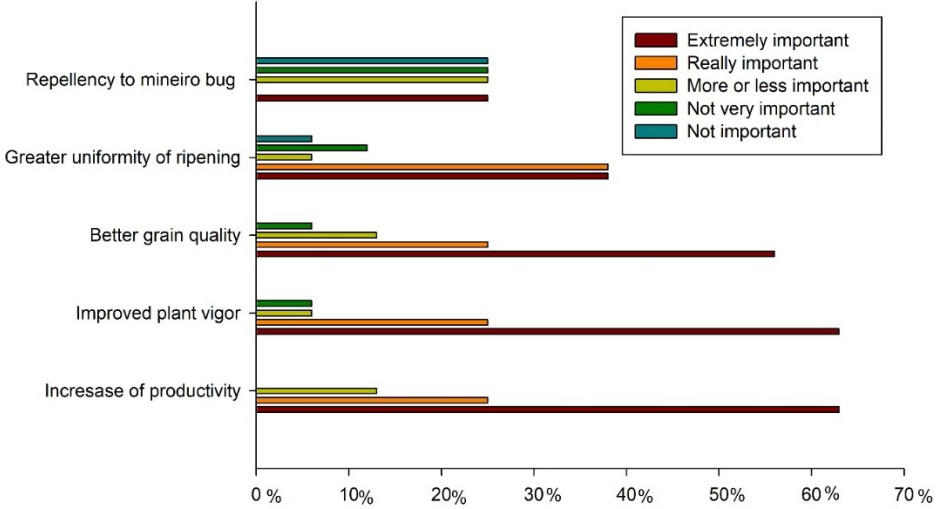

**Figure 8.** Evaluation of PKPF effects as measured by the coffee grower.

When questioned about the actions to be taken in the next ten years, in view of the expected rise in temperature, there was a consensus, even if at different levels, on the inclusion of the PKPF in coffee plantation management to mitigate potential losses (Figure 9). The vast majority (75%) disagreed, to some degree, with discontinuing the cultivation of coffee on their current properties, demonstrating that they are satisfied with growing coffee in the region. In fact, coffee farming is accordingly recognized as an activity that transcends generations of farmers.

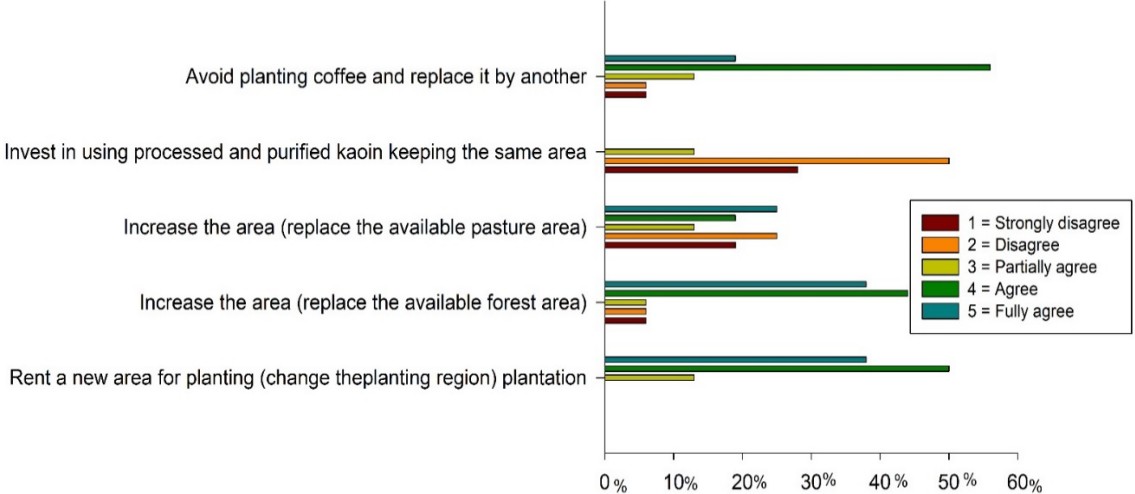

**Figure 9.** Future actions to maintain or increase productivity up to 2031 with predicted atmospheric air temperature rises.

Aware of the importance of caring for and preserving nature, 88% of the interviewees showed no interest in leasing new planting areas, 82% did not agree, to some degree, with the possibility of deforesting natural areas, and 57% agreed, to some degree, with converting pastures to coffee growing areas (Figure 9).

Brazil's leadership in world coffee production increasingly requires a correct attitude towards the environment, workers' rights, and product safety. By meeting the demands of the international market, and mindful of the commitments set forth in Agenda 2030 (UN), the Brazilian coffee industry must find new ways to manage coffee production, based on a set of technical guidelines to ensure the economic, social, and environmental sustainability of the coffee agribusiness.

Farmers from the Cerrado region of Minas Gerais can achieve this through Integrated Coffee Production Systems (ICP). The ICP is a standard that aims to insert coffee production into the modern concept of sustainable development, and can be used by all sectors of agribusiness interested in the certification of coffee production. It was developed based on several existing codes for coffee production and on the successful model proposed by the Ministry of Agriculture, Livestock, and Supply (MAPA) for Integrated Fruit Production (IFP), which was inspired by the IOBC (International Organization for Biological Control). The PIC, besides MAPA's official contribution, has the institutional and technological support of CBP&D/Café, which brings together 45 research institutions. This means that the producers, primarily in associations or cooperatives, will have better access to information and training opportunities, taking advantage of the specialized technical staff and infrastructure of the institutions belonging to the Consortium. The IFP advocates offering coffee with added social and ecological value and meets the requests of consumer countries that demand appropriate production conditions and product sustainability.

Considering all the beneficial effects resulting from PKPF applications and the fact that it is an inert and natural substance, this technology can be used in organic agriculture, and the commercial product is already registered and authorized in coffee growing and other crops in numerous countries. The rapid adoption and positive results obtained with

this technology in the countries where it has been introduced predict a strong impact on crop care, particularly in coffee, for which this work suggests PKPF should be included in the portfolio of good agronomic practices. This action can be included in the guidelines of technical standards for integrated coffee production (ICP-MAPA), as "recommended activities", in order to attenuate, and even mitigate, the harmful effects caused by high temperatures and excessive solar radiation.

## 5. Discussion

The study presents the state of the art for the use of PKPF technology in several crops and reports the commitment of the interviewed coffee growers with the preservation of the active longevity of the Cerrado biome. They declared and demonstrated, through concrete actions, their commitment to preserving the environment, their gratitude, and their reverence for mother earth, which guarantees sustenance for their families and others, contributing directly to the health and prosperity of the social fabric of the municipalities where they grow coffee. Examples of effective initiatives have been the replacement of synthetic chemical pesticides by bioproducts and organic inputs to protect against insect pests and diseases; the correct disposal of empty pesticide containers; the incorporation of crop residues and animal waste in the form of manure as alternative sources of nutrition for coffee crops, which can reduce the application of chemical fertilizers, reducing soil and groundwater contamination.

The occurrence of adverse climate events is a cause for concern, notably the rise in temperature, since temperatures above 23 °C interfere with the metabolism of the coffee plant, growth, flowering, and the production of coffee beans, due to the damage caused to the plant tissue, commonly known as "scalding". Of the group of coffee growers interviewed, 73% reported that they were unaware of the existence of any technology that could attenuate, and even mitigate, the harmful effects caused by high temperatures and excess solar radiation.

For leaves exposed to the sun, the benefits of using PKPF to mitigate against high temperatures were recognized by 94% of the coffee growers, and 86% affirmed that PKPF technology, when incorporated into production management, protects and improves the quality of the grains.

Concerning the possibility of PKPF as an environmentally accepted and technologically innovative input, that can increase the production and quality of coffee, it can be concluded that the use of processed and purified particle films, alone or in combination with foliar fertilizers and/or biostimulants, was confirmed by the farmers as an extremely important or very important practice. PKPF-based sunscreens represent a sustainable support tool for the coffee activity, even under the threat of increasing temperatures due to climate change.

The majority (75%) of coffee growers in the Cerrado region of Minas Gerais confirmed that PKPF did impact the environment, and all coffee producers plan, to some degree, to protect their crops by spraying this particulate film, to mitigate losses caused by increasing air temperatures.

A commitment by coffee growers to more environmentally correct crop management was observed. However, it should be noted that there is also some contradiction. For example, the majority of interviewed coffee growers (87%) produce coffee with the help of irrigation systems, but only 12.5% mention in the interview their concern with rational water use for irrigating crops. This result points to the opportunity for further research on the relationship between the use of water for crops and environmental impacts, in order to increase the commitment of farmers to the preservation of water resources.

With the increased demand for specialty, agro-ecological, and certified coffees, which require reduced chemical pesticide use, competitive pressures among suppliers, product traceability, and the growing presence of sustainability-related topics, there is a need to rethink habits and attitudes with the adoption of innovative and efficient practices, replacing conventional ones. This new trend in coffee growing through certification is a strategy to meet the global demands of strengthening environmental preservation and

ensuring food safety. This may be due to the fact that, in the last decades of the last century, there has been an accelerated devastation of forests, poor soil conservation, degradation of water resources, growing environmental pollution, and harmful effects on the environment, humans, and food caused by the excessive use of pesticides.

Brazil's leadership in the world coffee production increasingly requires a correct attitude towards the environment, workers, and product safety. By meeting the demands of the international market, and mindful of the commitments set forth in the Agenda 2030 (UN), the Brazilian coffee industry must find new ways to manage coffee production, based on a set of technical guidelines to ensure the economic, social, and environmental sustainability of the coffee agribusiness.

Farmers from the Cerrado region of Minas Gerais can achieve this through Integrated Coffee Production Systems (ICP). The ICP is a standard that aims to insert coffee production into the modern concept of sustainable development, and can be used by all sectors of agribusiness interested in the certification of coffee production. It was developed based on several existing codes for coffee production and on the successful model proposed by the Ministry of Agriculture, Livestock, and Supply (MAPA) for Integrated Fruit Production (IFP), which was inspired by the IOBC (International Organization for Biological Control). The ICP, besides MAPA's official contribution, has the institutional and technological support of CBP&D/Café, which brings together 45 research institutions. This means that the producers, primarily gathered in associations or cooperatives, will have better access to information and training opportunities, taking advantage of the specialized technical staff and infrastructure of the institutions belonging to the Consortium. The ICP advocates offering a coffee with added social and ecological value and meets the demands of consumer countries that are beginning to demand and value appropriate production conditions and product sustainability.

Considering all the beneficial effects produced by PKPF and the fact that it is an inert and natural substance, this technology is used in organic agriculture, and the commercial product is already registered and admitted in coffee growing and other crops in numerous countries. The rapid adoption and positive results obtained with this technology in the countries where it has been introduced predict a strong impact on crop care, particularly in coffee, and therefore this work suggests including the use of PKPF in the portfolio of good agronomic practices. This action can be included in the guidelines of technical standards for integrated coffee production (ICP-MAPA), as "recommended activities", in order to attenuate, and even mitigate, the harmful effects caused by the over-optimal temperature and excessive solar radiation.

Researchers reinforce that high temperatures, combined with water deficit, are the main climatic limitations to coffee production, which will lead to great losses for producers in the coming years, and the best strategy for the coffee growers to protect against droughts, which are normally exacerbated by high temperatures, is the sustainable management of the crop, which must be planned since its implementation, such as combining already known techniques with innovative methods, for example the use of PKPF, to mitigate the effects of climate change [2,3,71]. It is necessary to understand the effects of drought and high temperatures on the soil and on the coffee plants, proposing a combination of techniques such as: soil cover management (with organic material or even polyethylene film) and the ecological management of organic residues, among others [71].

**Author Contributions:** Conceptualization, N.d.M.R. and B.A.B.; methodology, N.d.M.R., B.A.B., R.M.L. and J.P.; software, N.d.M.R.; validation, N.d.M.R., P.R.d.S. and E.C.; formal analysis, N.d.M.R., P.R.d.S. and E.C.; investigation, N.d.M.R., D.P.d.A., P.R.d.S. and E.C.; resources, N.d.M.R. and D.P.d.A.; data curation, N.d.M.R.; writing—original draft preparation, N.d.M.R.; writing—review and editing, N.d.M.R., B.A.B., R.M.L., J.P., P.R.d.S. and E.C.; visualization, N.d.M.R.; supervision, B.A.B., R.M.L. and J.P.; project administration, N.d.M.R. and B.A.B.; funding acquisition, N.d.M.R. and B.A.B. All authors have read and agreed to the published version of the manuscript.

**Funding:** This research received funding by TKI/ NovaSource. This work was supported by CNPq fellowships PQ 303166/2019-3 for E.C.); Fundação Carlos Chagas de Apoio à Pesquisa do Estado do Rio de Janeiro (FAPERJ) grants (E-26/202.759/2018, E-26/210.309/2018, and E-26/210.037/2020 for E.C.; **E-26/204.220/2021** for P.R.d.S.); Pontifical Catholic University of Campinas for N.d.M.R.

**Institutional Review Board Statement:** Not applicable.

**Informed Consent Statement:** Not applicable.

**Data Availability Statement:** Not applicable.

**Conflicts of Interest:** The authors declare no conflict of interest.

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
