# Peer review of "The Advantages of Using Kaolin-Based Particle Films to Improve Coffee Production in the Minas Gerais Cerrado Biome"

_sustainability, doi:10.3390/su14084485_

Round 1
Reviewer 1 Report
The manuscript seeks to discuss the introduction of kaolin to increase the sustainability of coffee production in Brazil.
The first part of the manuscript discusses extensively the risks of coffee production related to climate change. In the second part, caoilino and its benefits in different agricultural contexts were presented. This two parts are quite good, well written and referenced. Then, the manuscript continues with a survey using sustainability questionnaires.
These two parts (climate change problem + caolin and sustainabilty questionaries) only partially go together and the manuscript is unorganized and, above all as a result, does not provide any scientific progress on the subject. The discussion of the results basically repeats the findings.
Finally a lot of style revisions are needed.
Author Response
The work was organized and better structured. The suggestion of another reviewer that adds to this one was also accepted, such as the synthesis of the state of the art (literature review) for a better structuring with the introduction.
Regarding the discussion of the results, the expectation that this work will present to the scientific community the importance of traditional and innovative sustainable production management and protection strategies in coffee farming, such as processed kaolin, adopted by producers in the Brazilian Cerrado in the context of family farming, which generally show resistance in terms of receptiveness to the diffusion and adoption of new technologies.
Reviewer 2 Report
The structure of the manuscript should be revised. Probably the title as well, or the content must be harmonised with it.
PKPF application should be justified in each topic of the Introduction section (not as "Review literature") which would make the content more coherent.
The use of abbreviations must be revised.
The numbering of the citations must be revised.
I indicated my comments and corrections in sticky notes in the pdf file of the manuscript (see attached).

Author Response
The work was organized and better structured. The suggestion of another reviewer was also accepted, in addition to this one, such as the synthesis of the state of the art (literature review) for a better structuring and alignment with the introduction.
The title, as well as the state of the art (literature review) have been modified and are in line with the objectives and conclusions, which would make the content more coherent.
The use of abbreviations was revised, as well as the number of citations.
Regarding the discussion of the results, the expectation that this work will present to the scientific community the importance of traditional and innovative sustainable production management and protection strategies in coffee farming, such as processed kaolin, adopted by producers in the Brazilian Cerrado in the context of family farming, which generally show resistance in terms of receptiveness to the diffusion and adoption of new technologies.
Reviewer 3 Report
The authors presented a typescript on the topic Sustainable coffee management in the cerrado of minas gerais using processed and purified kaolin-based particle film (PKPF).
In the introduction, the authors rightly provided information on the regions and conditions for growing coffee. The most important factors were distinguished, including temperature, average annual growth and geographical height. The authors also compared the cultivation of coffee in the sun and shade.
One of the most important research objectives was to evaluate the environmental sustainability practices adopted by farmers in the Carrado area of Minas Gerais.
I positively evaluate the research and the contribution of the authors' work to the creation of a very interesting publication on the development of coffee growing in the world.
Author Response
Yes, was one of the most important research objectives, to evaluate the environmental sustainability practices adopted by farmers in the Carrado area of Minas Gerais.
This article, all text has had an English revision.
Thanks for the contribution.

Round 2
Reviewer 1 Report
The manuscript was greatly and significantly improved compared to the previous version by incorporating almost all comments suggested by the reviewers.
Reviewer 2 Report
The manuscript was revised according to my suggestions.
The modified title of the paper expresses the content better.
I accept the new approach analysing the state of the art of the topic instead of the literature review. The shortened version is better too. Changing the titles of the subchapters simply justified the contents.
Revised Table 1 is more scientific in its recent form.
The extended discussion also provides more justification of the study and coherence with the content.
The abbreviations were corrected, though there are still some typos (spacing) in the text.